# Interdisciplinary Surgical Therapy of Extremity Soft-Tissue Sarcomas: A Personalized Resection and Reconstruction Algorithm

**DOI:** 10.3390/jpm13020262

**Published:** 2023-01-30

**Authors:** Justus Osterloh, Ingo Ludolph, Robert Grützmann, Alexander Meyer, Werner Lang, Raymund E. Horch, Katja Fechner, Andreas Arkudas

**Affiliations:** 1Department of Plastic and Hand Surgery and Laboratory for Tissue Engineering and Regenerative Medicine, University Hospital Erlangen, Friedrich-Alexander University Erlangen-Nürnberg (FAU), 91054 Erlangen, Bavaria, Germany; 2Department of Surgery, Comprehensive Cancer Center, University Hospital of Erlangen, Friedrich-Alexander-University Erlangen-Nürnberg (FAU), 91054 Erlangen, Bavaria, Germany; 3Department of Vascular Surgery, Comprehensive Cancer Center, University Hospital of Erlangen, Friedrich-Alexander-University Erlangen-Nürnberg (FAU), 91054 Erlangen, Bavaria, Germany

**Keywords:** extremity soft-tissue sarcomas, surgical resection, reconstructive surgery

## Abstract

Soft-tissue sarcomas (STS) are rare, but potentially life-threatening malignancies. STS can occur anywhere in the human body with the limbs being the most common site. Referral to a specialized sarcoma center is crucial to guarantee prompt and appropriate treatment. STS treatment strategies should be discussed in an interdisciplinary tumor board to involve expertise from all available resources, including an experienced reconstructive surgeon for an optimal outcome. In many cases, extensive resection is needed to achieve R0 resection, resulting in large defects after surgery. Hence, an evaluation of whether plastic reconstruction might be required is mandatory to avoid complications due to insufficient primary wound closure. In this retrospective observational study, we present data of patients with extremity STS treated at the Sarcoma Center, University Hospital Erlangen, in 2021. We found that complications were more frequent in patients who received secondary flap reconstruction after insufficient primary wound closure compared to patients who received primary flap reconstruction. Additionally, we propose an algorithm for an interdisciplinary surgical therapy of soft-tissue sarcomas regarding resection and reconstruction and present two problematic cases to emphasize the complexity of surgical sarcoma therapy.

## 1. Introduction

Soft-tissue sarcomas (STS) are malignant tumors deriving from mesenchymal tissue encompassing over 50 histological subtypes [1]. Although STS represent a rare condition with a proportion of <1% of all malignancies in adult patients [2], they are associated with a poor quality of life and poor prognosis for patients suffering from STS, primarily depending on the histological subtype [3]. The most common clinical presentation of patients with STS is a painless, enlarging mass which can arise throughout the body, with extremities being the most frequent localization, with a proportion of over 60% in adult patients [4]. About 75% of extremity STS occur in the lower extremities, whereas 25% arise in the upper extremities [4]. Due to their heterogeneity in evolution, appearance, and localization, there is poor evidence concerning diagnostics and treatment strategies in the literature compared to other malignancies. Advances in multidisciplinary treatment approaches have improved the care and outcome of patients with sarcoma. As the understanding of tumorgenesis and molecular patterns of sarcoma tissue has improved drastically over the last decades, radiotherapy (RT) and chemotherapy (CT) have become substantial treatment options. Nonetheless, surgical resection is the key element of treatment paradigms with R0 resection being an important prognostic factor and aim of the surgery [5]. Sarcomas typically become clinically present as unspecific painless swellings. Patients, as well as medical professionals, often underestimate the significance of a swelling of unknown primary. Therefore, diagnosis is frequently delayed and sarcomas become extensive in size before resection is performed. This highlights the need for plastic reconstruction of large defects after resection and emphasizes the importance of centralized care through a multidisciplinary team approach in a specialized center [6,7]. Studies with a high amount of evidence as well as clinical trials remain a challenge due to the rarity and heterogeneity of STS. In this retrospective observational study, we analyzed all patients with soft-tissue sarcoma of the extremities who were treated with surgical resection in the Sarcoma Center of the University Hospital Erlangen in 2021 to elucidate the issue of complications after resection and reconstruction. Furthermore, we aim to provide an algorithm for a multidisciplinary approach of the resection and reconstruction of extremity soft-tissue sarcomas (Figure 1).

## 2. Material and Methods

We performed a retrospective analysis of all patients who underwent interdisciplinary surgical therapy for extremity soft-tissue sarcoma at our University Hospital in 2021. A total of 31 patients were included. The inclusion criteria for this analysis were a histologically confirmed diagnosis of extremity STS and treatment at our Center. Patients with abdominal, retroperitoneal, thoracal and trunk soft-tissue sarcoma were excluded. Data including patient characteristics (median age and sex), treatment characteristics (neoadjuvant radiotherapy, neoadjuvant chemotherapy, adjuvant radiotherapy, adjuvant chemotherapy), sarcoma characteristics (tumor location, tumor size, sarcoma subtype), surgical treatment after tumor resection (e.g., primary wound closure, flap), follow up and complications after surgical therapy (in the subgroup of primary wound closure, primary flap reconstruction, secondary flap reconstruction) were extracted by reviewing digital medical records.

## 3. Results

In this retrospective study, we identified 31 patients with extremity soft-tissue sarcoma who received surgical resection at the Sarcoma Center of the University Hospital Erlangen in 2021. The median age of the included patients was 54.7 years, while 51.6% were male and 48.4% were female (Table 1). A total of 77.4% of the extremity STS were located in the lower extremities and 22.6% in the upper extremities. Seven patients (22.6%) received neoadjuvant radiotherapy and eight patients (25.8%) were treated with neoadjuvant chemotherapy. After surgery, nine patients (29%) received adjuvant radiotherapy and five patients (16.1%) received adjuvant chemotherapy. Tumor size varied from 1.5 cm to 17.5 cm in diameter resulting in a median tumor size of 6.5 cm. The most common STS subtype after histopathological analysis was undifferentiated pleomorphic sarcoma (10 patients, 32.2%), followed by leiomyosarcoma (four patients, 12.9%) and liposarcoma (four patients, 12.9%). Other diagnosed subtypes were synovial sarcoma, myxofibrosarcoma, myxoinflammatory fibroblastic sarcoma, chondrosarcoma and fibromyxoid sarcoma. Samples of three patients (9.7%) were categorized as unclassified sarcomas. 

After tumor resection, 21 patients (67.7%) were treated with primary wound closure after resection. Hence, complications (skin necrosis, seroma, would healing disturbance, bleeding, infection) occurred in six patients (28.6%) resulting in the need for secondary reconstruction with a flap in five (16.1%) cases. Out of the remaining ten patients (32.3%) without primary wound closure, seven (22.6%) received flap reconstruction, two (6.5%) were treated with negative pressure wound therapy (NPWT) and skin grafts, while one patient (3.2%) received only NPWT (Table 2). 

The most frequently used flap was a free vertical rectus abdominis myocutan (VRAM) flap (four patients), while two patients received a free anterior lateral thigh (ALT) flap. A free latissimus dorsi flap, pedicled VRAM flap, free tensor fascia lata (TFL), peroneus brevis flap and pedicled VRAM flap lateral intercostal artery perforator (LICAP) propeller flap was used for reconstruction in one patient each. No partial or complete flap losses were encountered. Two patient received negative pressure wound therapy (NPWT) followed by a skin graft, whereas one patient received NPWT only.

The median follow up was 12.03 months. Complications occurred in 6 out of 21 (28.6%) patients who received primary wound closure (infected seroma, wound healing disturbance and seroma, wound healing disturbance and hematoma, lymphedema, hematoma and infection, skin necrosis) resulting in secondary flap reconstruction in five cases. Additionally, complications occurred in 4 out of 12 (33.3%) patients who received flap reconstructions (seroma, wound healing disturbance, infection, bleeding). Three out of the four (75%) complications in the flap reconstruction subgroup occurred in patients who received secondary flap reconstruction after the failure of primary wound closure. In patients who received primary flap reconstruction, only one (1/7 = 14.3%) complication was reported (seroma).

## 4. Case Demonstrations

### 4.1. Case 1

A 79-year-old female patient was treated with a neoadjuvant radiotherapy and chemotherapy due to a diagnosed undifferentiated pleomorphic sarcoma in the right thigh (Figure 2). Afterwards, tumor resection and primary wound closure was performed. In the further course, skin necrosis and wound dehiscence occurred, therefore debridement and vacuum-assisted closure was performed. Afterwards, soft-tissue reconstruction was achieved by using a free vertical rectus abdominis myocutaneous flap anastomosed to a previously created arteriovenous loop. Three months after reconstruction, the patient presented to our hospital with wound healing disturbances at the medial border of the flap. Following this, defect reconstruction with debridement and transposition of the flap as well as the transplantation of a split skin graft from the contralateral thigh was performed. 

### 4.2. Case 2

A 47 year-old female patient presented with a dermatofibrosarcoma protuberans of the right groin (Figure 3). After resection, a vacuum-assisted closure was applied. Due to its high recurrence rate and small tumor free margins, a second resection was performed with soft-tissue reconstruction using a pedicled anterior lateral thigh flap and split thickness skin graft coverage of the ALT donor site.

## 5. Discussion 

In this study, we analyzed the importance of an interdisciplinary surgical treatment of extremity STS, including surgical resection and reconstruction, at our University Hospital Centre. This retrospective observational study shows the importance of an early interdisciplinary approach for every patient with an appropriate planning of resection and reconstruction. Referral to a sarcoma center should be initiated for any patient with a soft-tissue mass of unknown origin where STS are suspected. Treatment at a high-volume hospital is not only associated with a better prognosis for patients with STS [8], but it also enables the inclusion of those patients in clinical trials and studies of any kind to improve treatment and prognosis of patients suffering from STS in the future. 

After clinical examination, ultrasound examination provides the first impressions about the depth of invasion and size in diameter of a suspicious mass, but the gold standard for the diagnosis of STS is MR imaging with diffusion-weighted imaging. Typically, peritumoral postcontrast enhancement, heterogeneous T2 signal intensities and the presence of necrotic areas are signs of high grade (G3) tumors [9,10]. Additionally, MR imaging can reveal the size, depth of invasion and potential infiltration of the surrounding tissue of the tumor as well as local lymph node status, providing pivotal information for the resection, reconstruction and setting up of further treatment strategies in general. Furthermore, staging should be completed before biopsy. Our findings about the localization of STS are in line with the current literature where extremity STS are described to be more frequent in the lower extremities (75%) compared to the upper extremities (25%) [4]. In our study, we found that 77.4% of extremity STS were present it the lower and 22.6% in the upper extremity.

Depending on the size in diameter and depth of the mass, core needle biopsy, incisional or excisional biopsy should be performed. Fine-needle aspiration is not recommended because it only provides cytology and therefore lacks distinct information about the tissue architecture [11]. Excisional biopsies should only be performed in cases with superficial masses measuring ≤ 3 cm in diameter [11]. A recent meta-analysis demonstrated that core needle biopsy is not inferior to incisional biopsy in diagnosing the correct STS-subtype but is associated with fewer complication rates [11]. However, using multiple numbers of passes and a gauge size of ≥16 is recommended [11,12]. When using incisional biopsy for diagnostic purposes, drains should be placed in continuity with the skin incision or in the direct extension of the wound and a compressive dressing should be applied to prevent postoperative hematoma. 

Nonetheless, STS represent a large collection of different tumor subtypes with varying findings in histopathological analyses where the risk of error or false diagnosis is high. Out of the 31 patients presented in this study, eight different histopathological subtypes could be found. In three cases, an unclassified sarcoma was diagnosed (Table 1). In fact, the variability of STS diagnosis across pathologists is comparably high [13], underling the importance of an interpretation of the sample by an experienced sarcoma pathologist. In addition to a histopathological analysis, molecular testing of the obtained tissue was introduced to further investigate and elucidate tumor characteristics and subtype classification [14]. 

The planning of treatment strategies should be discussed in a multidisciplinary tumor board review within a specialized sarcoma team which should include a surgeon, a radiologist, a pathologist, a radiotherapist, and medical oncologist with expertise in the treatment of sarcoma. Radiotherapy and chemotherapy can be used as neoadjuvant or adjuvant regimes. This illustrates the importance of an interdisciplinary evaluation of each case at the time of diagnosis and before surgical resection.

Radiotherapy is well established in treatment regimens for stage II, III and IV STS with an increased risk of local recurrence compared to resection alone [15]. There is an ongoing debate about the ideal timing of radiotherapy. Recent studies were not able to demonstrate a superiority of neoadjuvant vs. adjuvant radiotherapy. Whereas neoadjuvant radiotherapy is associated with less edema and joint stiffness as well as decreased fibrosis after a long follow up, a greater risk of acute wound complications was found compared to adjuvant radiotherapy [16,17]. Systemic chemotherapy plays a role in the multidisciplinary disease management of patients with STS in a locally advanced or metastatic state. Cytotoxic chemotherapy, mainly consisting of anthracycline (doxorubicin) and ifosfamide derivates, has been the standard of care for the last few decades [18]. Nonetheless, the treatment outcome is poor, the grade of toxicity is high and recent trials have questioned the efficiency of the established chemotherapy regimens [19]. Additionally, a further understanding of molecular pathogenesis and the important pathways involved in STS leads to novel treatment approaches with targeted therapies. The field of chemotherapy in STS is now moving in the direction of histological subtype-dependent treatment strategies such as trabectedin for liposarcoma and leiomyosarcoma or eribulin for liposarcoma [20,21].

Depending on the histopathological results, the extent of surgical resection should be evaluated. After the pathological confirmation of STS, wide resection is necessary. In contrast to wide resection, so-called marginal resection is not sufficient, since a R0 resection cannot be guaranteed. Therefore, marginal resections should only be performed in cases where the histopathological examination of the biopsy shows no signs of malignancy. Historically, the amputation of the extremity was state of the art in the treatment of sarcomas of the upper and lower extremities. Through the years, it was shown that the amputation of the extremity is not advantageous regarding long-term survival compared to R0 resection [22,23,24]. Important is the so-called wide resection in the healthy tissue surrounding the tumor with tumor free margins [25,26]. Ideally, the tumor is not visible during the resection. Anatomical borders such as fascia or periost are also important during the resection. Additionally, resection of the entire compartment is not necessary when the tumor does not invade the origin and insertion of the muscles. The resection of adjoining muscle tissue surrounding the tumor results in the same long-term survival as the resection of the entire compartment, leading to the preservation of function when possible, combined with radical tumor resection. Additionally, adjoining nerves can be preserved without risking inadequate tumor resection when epineural dissection is performed [27]. Tumor infiltration of vessels and/or nerves leads to resection and when necessary, the reconstruction of these structures, but not inevitably to amputation of the extremity. Nevertheless, there are also indications for limb amputation, especially in cases of massive tumor infiltration or when complex reconstructions are not possible due to age or poor general condition [28]. Overall, surgical therapy can be divided into oncological resection and surgical reconstruction of the tissue defect (bone and/or soft tissue) and nerve/vessel or functional reconstruction [29,30]. 

The reconstruction itself can be divided into primary wound closure and functional reconstruction. Primary closure attempts should only be performed when a tension free closure can be guaranteed and after an evaluation of aggravating factors for wound healing such as irradiation. In all other cases, defect reconstruction is indicated, to prevent complications due to insufficient skin closure. In our study, 6/21 patients developed complications after primary wound closure and five out of those six patients needed secondary flap reconstruction after the primary wound turned out to be insufficient. 

In total, complications occurred in 4 out of 12 (33.3%) flap reconstructions (seroma, wound healing disturbance, infection, bleeding). It must be mentioned that three out of the four cases with complications in the flap reconstruction group were patients who received secondary flap reconstruction after the failure of primary wound closure. In the subgroup of patients who received primary flap reconstruction, only one (1/7) complication occurred (Table 3). 

This highlights the importance of an interdisciplinary surgical assessment of every patient, including an evaluation from an experienced reconstructive surgeon concerning the suitability for primary wound closure to avoid complications and unnecessary secondary surgeries. Additionally, the diversity of the flaps used for reconstruction underlines the complexity of the surgical treatment of extremity STS (Table 2). Depending on the localization, diameter, depth, and surrounding vessels of the defect as well as the individual condition of every patient, the ideal flap should be evaluated carefully.

Among various complications, postoperative lymphedema and lymphoceles are major concerns after STS resection and reconstruction [31,32]. After extensive resection, significant damage to the lymphatic pathways is frequent. Therefore, besides defect reconstruction, surgeons should consider lymphatic restauration as well, to provide sufficient lymph drainage and to prevent lymphatic complications. The most used procedure is lymphatic restauration with prophylactic lymphaticovenous anastomoses (LVA) at the time of reconstructive surgery [33]. However, other strategies have been proposed lately. The usage of lymph-interpositional-flap transfer (LIFT) or lymphatic flow-through flaps (LyFT) is gaining popularity [33,34,35,36]. Lymphatic interpositional flaps preserve the lymphatic system from the donor to recipient site, where neolymphangiogenesis regenerates lymphatic drainage [34]. The system behind lymphatic flow-through flaps is a combination of the reconstruction of tissue defects and lympho-venous anastomoses between leaking vessels of the donor site and superficial flaps of the transplanted flap as a lymphatic derivation concept [37]. Pre- and intraoperative indocyanine-green (ICG) lymphangiography is also helpful to visualize lymphatic vessels in any procedure for lymphatic restauration [38,39].

Regarding the timing of reconstruction, either a one-step procedure throughout the tumor resection surgery or alternatively a delayed reconstruction after the histological confirmation of an R0 resection can be performed. One-step procedures offer the advantage of reduced hospitalization time and early rehabilitation, whereas two-step procedures can prevent complications due to positive margins after initial resection and enable the more precise planning of the required reconstruction. 

However, in cases where complications seem more likely, a two-step procedure should be indicated. Current data supports the advantages of a temporary, vacuum-assisted closure regimen (negative pressure wound therapy, NPWT) between resection and definitive reconstruction, leading to a lower risk of wound complications [40]. However, one should evaluate the vessel status of the extremities, especially around the tumor region, before planning surgical regimens for each individual patient. Therefore, CT angiography can be used preoperatively to provide detailed information about the vessel anatomy [41]. Moreover, imaging techniques such as indocyanine green angiography or blood flow analysis can be used intraoperatively to further ameliorate reconstruction safety and prevent flap loss [42,43]. 

In some cases, where free flap reconstruction is necessary and the only large vessels in the region of the tumor are invading the tumor itself, one-step procedures should be considered because the vessels would otherwise be ligated during the resection surgery, leaving no recipient vessels for free flap reconstruction. Alternatively, arteriovenous loops can be an effective tool to provide recipient vessels in the defect zone after resection in two-stage procedures. 

Resected large nerves and vessels can be reconstructed using nerve (mostly sural nerve) and vein (mostly great saphenous vein) grafts, although in some cases tendon transfers should be prioritized when motoric nerves are resected, especially in older patients. Additionally, functional reconstruction of resected muscle tissue is mainly performed using tendon transfers when possible, whereas free functional muscle transfers are rare [44]. Compared to this, soft-tissue reconstruction can be achieved using local or free flaps. Due to the increased safety of microsurgically transplanted free flaps, this type of flap has been used more commonly, especially when large tumors are resected [45]. In some cases, after tumor resection, no recipient vessels are present. Therefore, an arteriovenous loop is created in a one- or two-step procedure which can be then used as a recipient vessel in the further course of treatment. Additionally, local pedicled flaps can be used for defect coverage such as the pedicled anterior lateral thigh (ALT) flap or the pedicled vertical rectus abdominis myocutaneous (VRAM) flap for groin defects [43,46,47]. These flaps can also be transplanted in a free flap manner [48,49]. In larger defects, a latissimus dorsi flap can be used, in combination with a serratus anterior muscle flap or a parascapular flap to expand the flap area [50,51].

The validation of the presented study is limited by its small sample size. Therefore, further investigations are necessary to confirm the observed trends, which highlight the role of an early evaluation of patients for reconstructive surgery to avoid complications due to insufficient primary wound closure. 

## 6. Conclusions

Because of the heterogeneity and rarity of STS, disease management and treatment should be performed by an experienced multidisciplinary team at a specialized center from diagnosis to follow up. Radical oncological resections frequently require plastic reconstruction to provide limb salvage and preservation of function. This manuscript highlights the importance of an early interdisciplinary surgical evaluation of every patient to consider if plastic reconstruction might be necessary in order to avoid complications due to insufficient primary wound closure after resection. 

## Figures and Tables

**Figure 1 jpm-13-00262-f001:**
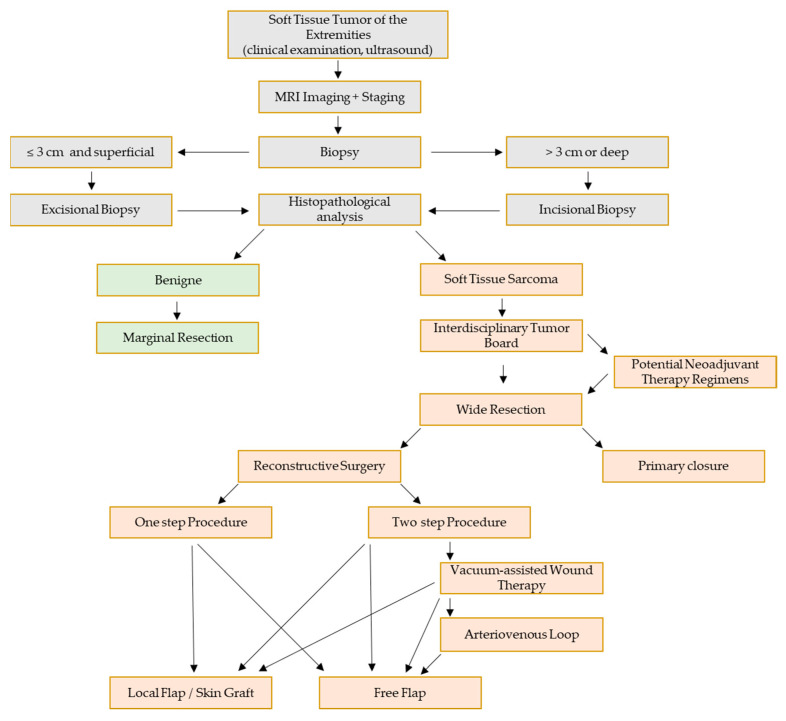
Proposed therapy algorithm for interdisciplinary surgical therapy of extremity soft-tissue sarcomas.

**Figure 2 jpm-13-00262-f002:**
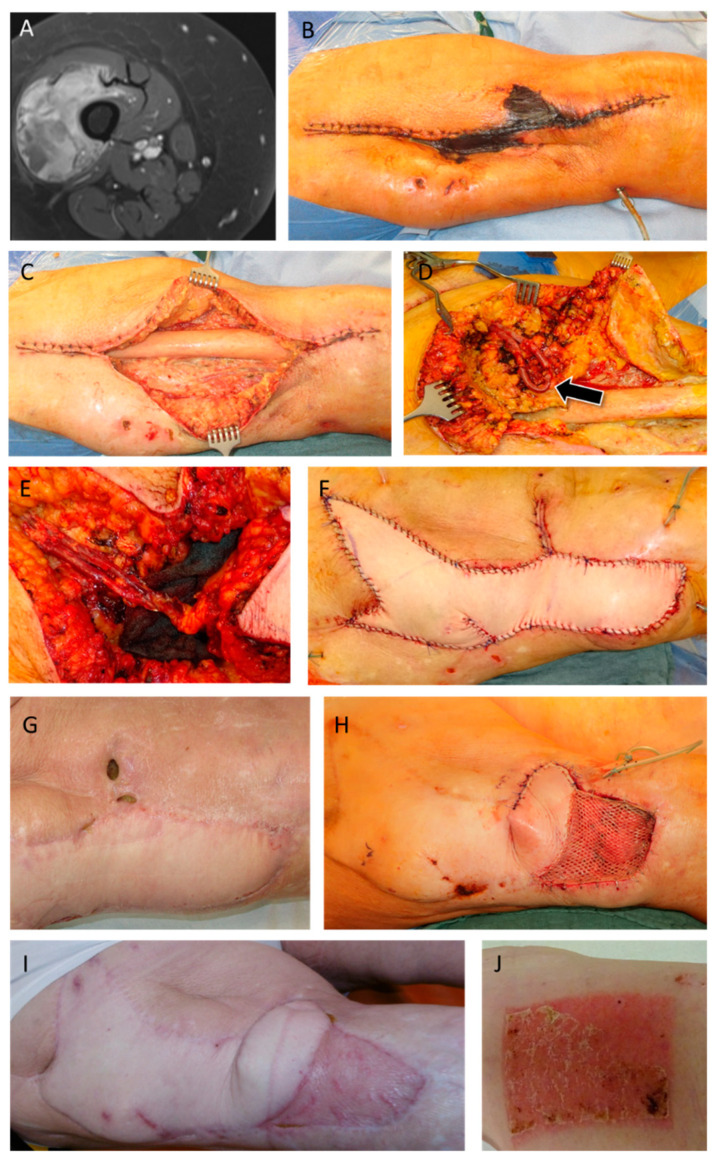
MRI of a 79 year-old female patient reveals an undifferentiated pleomorphic sarcoma in the right thigh (**A**). After resection and primary wound closure, skin necrosis occurred (**B**). After debridement (**C**) an arteriovenous loop was created (**D**, arrow) and a VRAM flap was transplanted microsurgically and anastomosed to the AV Loop pedicle (**D**–**F**). After occurrence of wound healing disturbances at the medial border of the flap (**G**), a transposition of the distal flap and split skin grafting was performed (**H**). Postoperative clinical presentation after revision showing stable wound conditions (**I**,**J**).

**Figure 3 jpm-13-00262-f003:**
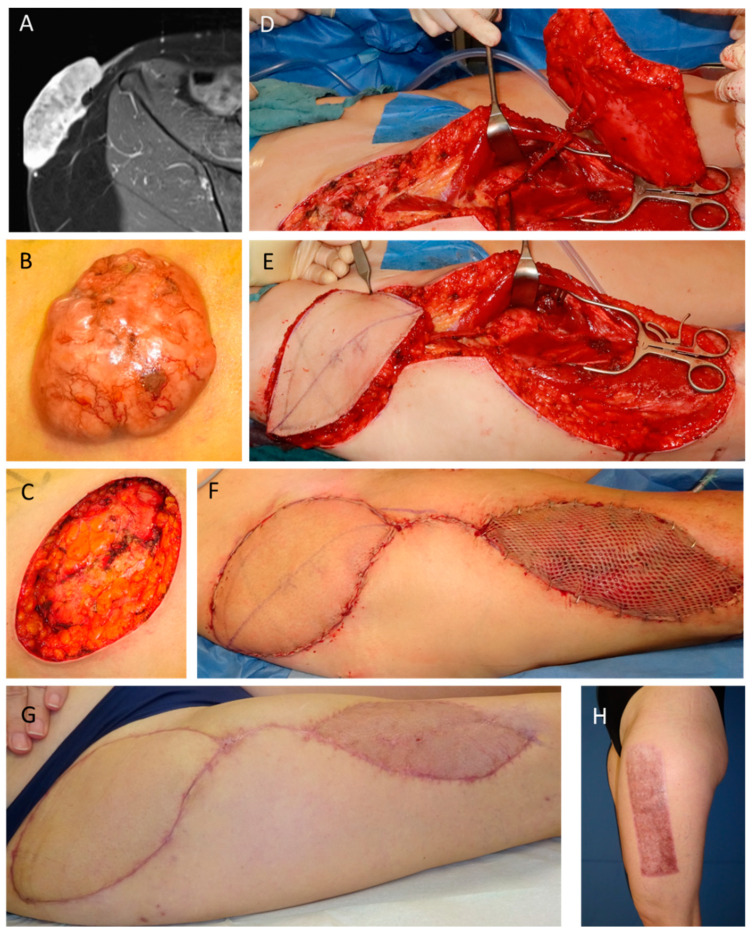
**A** 47 year-old female patient with a dermatofibrosarcoma protuberans of the right groin (**A**: MRI, **B**: clinical presentation). The tumor resection led to a large soft-tissue defect (**C**), which was covered using a pedicled ALT flap (**D**: flap pedicle (descending branch of the lateral circumflex femoral vessels), **E**: flap insertion, **F**: final result with split thickness skin graft coverage of the ALT donor site). Clinical presentation 2 months after reconstructive surgery shows sufficient defect reconstruction (**G**,**H**).

**Table 1 jpm-13-00262-t001:** Characteristics of the 31 included patients (Sarcoma Center, University Hospital Erlangen) including Age, Sex, Radiotherapy (RT), Chemotherapy (CT), Localization, Tumor size, Soft tissue sarcoma (STS) Subtype and Follow up.

Patient/Treatment Characteristics	Values/Numbers		
Median Age (Years)	54.7 (20–86)		
Sex	Male	16	51.6%
	Female	15	48.4%%
Neoadjuvant RT	Yes	7	22.6%
	No	24	77.4%
Neoadjuvant CT	Yes	8	25.8%
	No	23	74.2%
Adjuvant RT	Yes	9	29.0%
	No	22	71.0%
Adjuvant CT	Yes	5	16.1%
	No	26	83.9%
Localization	Upper Extremity	7	22.6%
	Lower Extremity	24	77.4%
Tumor size (cm, *n* = 23)	6.5 (1.5–17.5)		
STS Subtype	Undifferentiated pleomorphic sarcoma	10	32.2%
	Leiomyosarcoma	4	12.9%
	Liposarcoma	4	12.9%
	Synovial sarcoma	3	9.7%
	Myxofibrosarcoma	3	9.7%
	Myxoinflammatory fibroblastic sarcoma	2	6.5%
	Chondrosarcoma	1	3.2%
	Fibromyxoid sarcoma	1	3.2%
	Other unclassified sarcoma	3	9.7%
Follow up (Month)	12.03 (0–22)		

**Table 2 jpm-13-00262-t002:** Surgical Treatment after Extremity Soft-tissue Sarcoma Resection in 2021 (Sarcoma Center, University Hospital Erlangen).

Surgical Treatment after Tumor Resection (*n* = 31)	Procedure	Flap
Primary Wound Closure (*n* = 21)	Secondary Flap Reconstruction (*n* = 5)	Free VRAM Flap (*n* = 3) Free Lat. Dorsi Flap (*n* = 1) Pedicled VRAM Flap (*n* = 1)
No Primary Wound Closure (*n* = 10)	Primary Flap Reconstruction (*n* = 7)	Free ALT Flap (*n* = 2) Free TFL Flap (*n* = 1) Free VRAM Flap (*n* = 1) Peroneus Brevis Flap (*n* = 1) Pedicled VRAM Flap (*n* = 1) LICAP Propeller Flap (*n* = 1)
	NPWT + Skin Graft (*n* = 2)	
	NPWT without Reconstruction (*n* = 1)	

**Table 3 jpm-13-00262-t003:** Complications after surgical therapy of extremity STS.

Subgroup	Complications
Primary Wound Closure (*n* = 21)	Infected Seroma (*n* = 1) Wound Healing Disturbance and Seroma (*n* = 1) Wound Healing Disturbance and Hematoma (*n* = 1) Lymphedema (*n* = 1) Hematoma and Infection (*n* = 1) Skin Necrosis (*n* = 1)
Primary Flap Reconstruction (*n* = 7)	Seroma (*n* = 1)
Secondary Flap Reconstruction (*n* = 5)	Wound Healing Disturbance (*n* = 1) Seroma (*n* = 1) Wound Healing Disturbance and Infection (*n* = 1)

## Data Availability

Not applicable.

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
