# Peer review of "Interdisciplinary Surgical Therapy of Extremity Soft-Tissue Sarcomas: A Personalized Resection and Reconstruction Algorithm"

_jpm, 2023, doi:10.3390/jpm13020262_

Round 1

Reviewer 1 Report

Congratulations to the authors for their comprehensive and well-written article.

The reviewed submission is a complete article on soft tissue sarcomas, presenting in a condensed form the most relevant information on clinical presentation, diagnosis, prognosis, and treatment available nowadays. Furthermore, the authors include a unique treatment algorithm that I find very interesting and some cases from their personal experience.

I see no flaws in the article in its present form. I have only one question regarding the approval of the institution's Ethical Committee - since clinical cases are presented. Does it exist? If it does, the authors should mention that in the paper.

Reviewer 2 Report

This is a clinical study on multidisciplinary approach to extremity soft tissue sarcoma including resection and reconstruction. The study included 31 patients who underwent sarcoma treatments in a University hospital in 2021, and described the results in a non-systematic review manner. Although the manuscript is of clinical significance potentially worth shared among medical staffs, the following points should be revised before considering publication;

Study Design

1. The design and methodology of the study are unclear, and manuscript writing is inappropriate for an original article; the current manuscript looks rather a review article. If the authors want to publish the manuscript as an original article, the entire manuscript should be revised to fit as an original article of a retrospective observational study.

Abstract

2. Please clearly describe the background and the aim of the study, the study design, the methodology, the objective results, and conclusions in the Abstract.

Methodology

3. The study design and methodology are unclear in the current manuscript. Currently, the patients included in the study is described in the Introduction, which should be described in the Results; inclusion/exclusion criteria should be described in the Methods. Please clearly describe primary/secondary outcomes of the study in the Methods/Results.

4. The study include patients undergoing sarcoma treatments only in 2021. It is much better to extend the study period for far better analyses.

5. Please add Figures showing long-term postoperative results of reconstructed site and donor site.

Discussion

6. Discussion is combinedly described in the sections of methodology. Discussion should be described in a separate section, and description of discussion in the current methodology sections should be moved to the Discussion; only study methods/results should be described in the methodology sections.

7. Quality of life of sarcoma survivor is becoming a big issue in sarcoma management. Among various complications, postoperative lymphedema is one of the most significant sequelae following sarcoma resection and reconstruction. Increasing number of studies reports usefulness of lymphatic reconstruction in sarcoma surgery to prevent lymphedema.  It is strongly recommended to discuss lymphedema prevention in sarcoma treatment with the following citations; PMID 33867280, 32964444, 35056375, 29965920, 36226524

Conclusions

8. Please describe only the facts clarified by the study in the Conclusions. The current Conclusion is too long, and should be shortened up to 3 sentences.
